# A Wind-Solar Hybrid Energy Harvesting Approach Based on Wind-Induced Vibration Structure Applied in Smart Agriculture

**DOI:** 10.3390/mi14010058

**Published:** 2022-12-26

**Authors:** Lili Xia, Shaofei Ma, Peng Tao, Wanpeng Pei, Yong Liu, Lei Tao, Yipeng Wu

**Affiliations:** 1Jiangsu Agricultural Machinery Experimental Identification Station, Nanjing 210017, China; 2State Key Laboratory of Mechanics and Control of Mechanical Structures, Nanjing University of Aeronautics and Astronautics, Nanjing 210016, China

**Keywords:** photovoltaic electricity generation, wind-induced vibration, energy harvesting, smart agriculture

## Abstract

Solar energy harvesting devices are widely used in smart agriculture nowadays. However, when lighting conditions are weak, such as through the night or on cloudy days, efficiency decays a lot. Additionally, as time goes by, more and more dust and bird droppings accumulate on the panel, which decreases the performance significantly. This paper aims to overcome the disadvantages mentioned above, and a novel wind–solar hybrid energy harvesting approach is proposed with an oscillation-induced dust-cleaning function. A wind-induced vibration device is specially designed in order to generate electrical energy and/or clean the photovoltaic panel. While in good lighting conditions, the device could keep the panel in a stable state and optimize the photovoltaic power generation efficiency. Such a hybrid energy harvesting approach is called a “suppress vibration and fill vacancy” algorithm. The experimental platform of the proposed device is introduced, and both experimental and simulation results are attained, which prove that using this device, we could realize multiple purposes at the same time.

## 1. Introduction

China has been a big agricultural country since ancient times; China’s economy is inseparable from the development of agriculture. With the development of socialist modernization, many young people in rural areas go to cities for work, which causes the problem of an insufficient workforce [1]. The willingness of peasants to grow grain has become weaker, and arable land abandonment is very common now, which poses the issue to the countryside of who will plant the land and how to plant the land [2]. At the same time, the business space of agriculture is increasing rapidly, as more and more peasants choose to rent their land to a small number of large grain growers, organizations, and enterprises, so that they realize the unified scale operation of agriculture. All of these situations challenge the management of farmland and crops. The ways of getting information and environmental parameters for peasants is outdated, and they lack scientific instruction too. As a result, these factors are adverse to the improvement of agriculture efficiency and crops yield.

With the development of big data, IoT, and sensors, information and communications technology, simplified as ICT, has begun to integrate with agriculture [3]. In consequence, smart agriculture, characterized by intelligence, networking, and digitalization, has appeared and become the new direction of modern agriculture development. It combines contemporary technology with agriculture and avoids the issues of pollution, soil degradation, etc. Developed countries started the research on smart agriculture in the 1980s, especially in America, and many scientific companies providing smart agriculture solutions have been established over the past twenty years. Antonacci et al. [4] demonstrated that the application of nanotechnology in smart agriculture has the effect of improving the pollution caused by agricultural environments. Cicioğlu and Çalhan [5] aimed to achieve productive corn harvest in large-scale fields with the help of IoTs hardware and software facilities. Their system used heterogeneous sensor nodes which were capable of sensing the acoustic, rain, wind, light, temperature, and pH levels of the cornfields for smart agriculture applications. Monsanto in the United States began researching smart agriculture as early as 2001, allowing farmers to use the Internet to track the overall condition of their crops more efficiently and quickly [6]. Compared with these developed countries, China’s research and development on smart agriculture started relatively late. Until 2014, the government of China proposed the definition of “intelligent agriculture”. However, with the encouragement of national policies, many universities and institutes in China participate in the research of smart agriculture-related technologies. Tao et al. [7] conducted a review of the internet of things communication technologies in smart agriculture and their challenges. Wang et al. [8] proposed a hybridized energy harvesting device based on a high-performance triboelectric nanogenerator for smart agriculture applications. Gui et al. [9] proposed a self-powered, smart agriculture real-time sensing device based on a hybrid wind energy harvesting triboelectric-electromagnetic nanogenerator. Among these systems, sensors play a very important role and are widely used. A sensor is a kind of device that could translate a specific signal (physical, chemical, or biological signals, etc.) into a signal which is easy to measure and detect according to a certain law. Sensors are usually composed of sensitive components, converter components, and a converting circuit. According to their usage, we divide the sensors into several types, such as voltage, current, temperature, etc. With these sensors, we could realize at least three functions in the outdoor crop environment; they are monitoring temperature, smart irrigation, and locating spraying pesticides drone, etc.

Nowadays, many smart agriculture solutions are just applied in greenhouses where vegetables and fruits are grown. There are still relatively few cases of smart agriculture systems for the large-scale outdoor growth of crops. In addition, we face the problem of a lack of electric energy, which means it is very difficult for us to provide energy for sensors working in outdoor environments. As a result, this article proposes a new type of wind–solar complementary energy harvesting device based on the environmental energy self-harvesting technology.

## 2. Self-Powered Wireless Sensor and Environment Energy Sources

Smart agriculture is a collective term for digital agriculture, precision agriculture, etc., and is the trend and goal of modern agricultural development. Smart agriculture integrates modern science and technology with agricultural production, especially along with the application of wireless sensor network technology. It is predicted that the world population will rise to 9.2 billion in 2050, and smart agriculture will be a key technology to solving the problem of food shortage.

With the help of wireless sensors, traditional agricultural production and operation becomes automatic, unmanned, and intelligent [10]. Some developed countries have formed a relatively mature smart agriculture solution and a relatively complete industrial system [11]. Agricultural production equipment could be automatically controlled according to the environmental parameters of crops. As a result, crops would not be affected by climatic conditions, making them grow in the best ecological environment to improve crop yield. However, the issue of how to obtain the energy needed to sustain the operation of information-sensing devices still puzzles researchers. For wireless sensors, if the traditional method of battery-powered technology is adopted, the maintenance cost for changing the batteries frequently will be substantial, and the number of deserted batteries will be a heavy burden for the environment.

Self-powered IoT technology [12,13] seems to provide a solution for power supply. This technology refers to sensing devices that are passive in the network, meaning not equipped with batteries or connected to a power cable to provide energy, which means that the sensing devices operate independently and automatically. As a result, the required energy could only be acquired from its surroundings. Luckily, as the power consumption of the devices continues to decrease, this type of environmental energy self-harvesting technology becomes feasible.

Figure 1 shows the structural chart of a typical autonomous sensing device (right side) and an operation principle of an energy conversion device (right side), taking an agriculture environment as an example; the powered energy mainly comes from solar and wind. Wind and solar energy in the environment around the sensor node are harvested through the energy conversion device and are converted into electricity. After that, a stable direct current is output through the power management unit (PMU). On the left side of the figure is the actual device equipped with smart sensors in agriculture. It collects physical information from the environment through sensors and is intelligently processed by a microprocessor. The necessary data information is then uploaded to the base station in the smart agriculture system through the radio frequency module. This article mainly discusses the harvesting of solar energy and wind energy from the environment on the scene of power equipment operation.

To make sure that self-harvesting energy technology operates stably and reliably, it is necessary to consider the source of solar and wind energy under a specific environment. Accordingly, we chose the working tower micro weather sensors in Nanjing and deduced the data. The sensors were set up on the transmission line tower in an agricultural–electric environment surrounded by crops in Jiang Ning District, City of Nanjing. Using the data, the relationship chart between the parameters (light intensity and wind speed) and the date could be attained.

The maximum value of sunlight intensity each day with respect to the corresponding date is shown in Figure 2. As the working sensor does not have 24 h data and given the different weather conditions, the fluctuation of light intensity is obvious. However, in the general trend, it could be seen that after one year and a half of micro meteorological sensor using, albeit in the summertime, the measured light intensity becomes smaller overall; the maximum value is 400 W/m^2^, which is 2.3 times less than the data from the winter of 2019. This phenomenon is the result of the sensitive element in the sensor being covered by dust or other opaque things which it could not clean off by itself. As a result, the measured data has relatively big errors. These problems will occur in the solar energy harvesting device too, that is the photovoltaic (PV) panels will also be covered by bird droppings, dust, etc. If regular maintenance cannot be executed, then the efficiency of power generation will continue decreasing. Some documents report that under extreme conditions, the efficiency of PV power decreases by 51% in half a year.

In Figure 3, the same sensor is used to calculate the average wind speed with respect to date and it yields a diagram of changing relationship. It can be seen that the tower environment has wind in every season, and the range of wind speed is from 0.5 m/s to 5 m/s, which is not a dramatic change. Furthermore, most of the time, the average wind speed changes between 1 m/s and 3 m/s. In consequence, wind energy is harvested in our proposed method to offset the loss of solar energy by analyzing the features of wind speed.

## 3. Principles of the Novel Wind–Solar Hybrid Energy Harvesting Approach

### 3.1. Operation Principles

Normally, the energy density of solar energy is two or three orders of magnitude higher than other types in daytime under outdoor conditions, and this technology for energy conversion is relatively mature. In consequence, we choose PV panels based on the PV effect, which would enable the harvest solar energy efficiently to generate electricity. This is the easiest way to implement environmental energy harvesting. However, at night, on cloudy, rainy days, and under poor light conditions, it is very difficult to harvest enough solar energy. In addition, there is efficiency decay in PV devices due to accumulated bird droppings, dust, etc., provided that no cleaning is applied to those devices.

For micro-power wind energy harvesting, energy conversion is performed based on the law of electromagnetic induction, and it requires a rotating component, in case it is unnecessary for maintenance during its life cycle. As a result, big money is spent spent to manufacture highly reliable bearings, blades, and other structural parts. As a result, for small-scale wind energy harvesting technology, more and more researchers are studying a wind-induced vibration structure [14,15,16]. Namely, there is a trend of transferring wind energy to vibration energy first, and then converting vibration energy to electrical energy through the mechatronic conversion element [17].

For the practical application environment of smart agriculture, wind–solar hybrid energy harvesting technology is a good choice to improve the generation power and the survivability of self-powered devices [18]. In this case, integrating the wind-induced vibration device and PV panels on a single unit could improve the integration of the device. Namely, micro-power PV panels would be integrated with the blunt body which is shown in Figure 4.

However, when applying the mentioned integration method, the PV panel oscillates constantly. Therefore, there is the question of whether the oscillation will affect the performance of the PV power generation occurs. Based on this, a method of oscillation simulations is introduced in order to test a piece of PV panel, which is shown in Figure 5.

PV power generation generally requires PV panels to be static, that is, static with respect to the earth. Even PV panels with a sun-setting tracking system are in quasi- static conditions, so PV power generation efficiency is relatively high. However, for the wind induced vibration–solar hybrid energy harvesting device proposed in this article, when the PV power generation is at the dominant place, there are chances of wind-induced oscillation. Based on this, Figure 5 sets up the performance test platform under the dynamic conditions of micro-power PV panel, the chosen panel was a micro solar epoxy plate with a size of 54 × 54 mm^2^, and it was attached to a phone stepper in order to simulate mechanical oscillations. Additionally, an external programmable resistor board was attached, and the whole device was placed in sunlight for observation.

To quantitatively compare the output performance of PV panels, Figure 6 gives the curve of the solar panel output power with respect to load resistance under static and oscillating conditions, respectively, at noon on 11 March 2022. On that day, the weather was sunny, so the overall output power was a little high, up to 140 mW or higher. Comparing the static curve with the oscillating one, it could be found that, under static conditions, it had better output power performance in the range of load resistance. This conclusion clearly shows that the performance of the PV plane strongly depends on its fixed state, which means that the integration of the PV panel and the blunt body would decrease the power generation efficiency on sunny days (the reduction usually being around 10% to 15%), and it therefore needs a new method of wind–solar hybrid energy harvesting to overcome this problem.

In order to solve the problem mentioned above, a new type of wind–solar hybrid energy harvesting technique is proposed. As shown in Figure 7, the traditional hybrid harvesting approach is to capture wind and solar energy, respectively, and then a complicated PMU is used to store the generated electrical energy so that a DC voltage is provided.

In this proposed approach, when solar energy is in a dominate place, wind power is not harvested, instead using a little electricity to suppress vibration in order to improve the power generation efficiency of the solar energy. When the power generation efficiency of solar power is very low, then the device will harvest energy and clean the dust from the PV plane at the same time. This method integrates a PV panel and wind-induced vibration device in one structure, so that when the environmental wind speed exceeds the threshold, substantial vibration will occur in the structure. This vibration will bring two favorable benefits. One is that substantial vibration could generate power through vibration energy harvesting technology and the overall output is relatively smooth. Another is that the significant wind-induced oscillation vibration of the structure along with rainwater could clean the dust in solar panels effectively, and therefore there is no need to maintain the panels regularly. 

Although little electrical energy is utilized to suppress the oscillation of the PV plane, this process improves the stability of the PV plane, thus enhancing the PV power generation quality. The benefits of the improved generation power are far greater than the consumption of the controlled power. When wind power generation is mainly being used or the solar panels need to be cleaned, then the piezoelectric element will be used to gather vibration energy, and the structure will be in the state of large oscillation at the same time, which is the best choice. For micro-power PV electricity generation devices, the above hybrid energy harvesting control algorithm is called the “suppress vibration and fill vacancy” algorithm, as shown in Figure 7.

The principles of solar energy harvesting are based upon the PV effect, which is the phenomenon that when a semiconductor or the place where a semiconductor is combined with metal is exposed to light or other electromagnetic radiation, it will generate voltage and current. Generally speaking, the PV effect element outputs direct current, and the output power is positively correlated with the light intensity. In the energy self-harvesting sensing system, the electrical energy obtained by the PV effect first enters the power storage module through the charging circuit, and then the device provides electricity to the subsequent device.

In order to realize the “suppress vibration and fill vacancy” algorithm, the key technology is the energy harvesting interface circuit connecting the piezoelectric element; through it, the bidirectional control of piezoelectric energy can be realized. When a little electrical energy is injected into the piezoelectric element, then piezoelectric element could suppress the vibration of the structure and make the solar panels in a relatively stable state, which brings more energy gains. When energy is harvested from piezoelectric element, the vortex-induced vibration mechanism maintains a certain oscillation

### 3.2. Implementation of the Proposed Technique

In order to realize the “suppress vibration and fill vacancy” algorithm, the key technology is the energy harvesting interface circuit connecting the piezoelectric element through it the bidirectional control of piezoelectric energy could be realized [19]. If a little electrical energy is injected into the piezoelectric element, then the piezoelectric element could suppress the vibration of the structure and put the solar panels into a relatively stable state, which results in more energy gains. When the energy is harvested from piezoelectric element, the vortex-induced vibration mechanism maintains a certain oscillation amplitude. This could convert wind energy into electricity and clean the dust on the panels at the same time.

The piezoelectric energy interface circuit based on a transformer could realize the functions mentioned above. As shown in Figure 8, the interface circuit combines an optimized synchronous electric charge extraction (OSECE) circuit [20] and a synchronized switch damping based on energy injection (SSDEI) circuit [21]. On the basis of a flyback transformer, energy could be exchanged on the primary and secondary sides of the transformer, which realizes the two-way control of energy. This means that the electric energy generated by wind-induced vibration could be harvested through the OSECE circuit, and the vibration of the structure could be suppressed through the SSDEI circuit. Equations (1) and (2) express the normalized harvesting energy [20] and the vibration attenuation [21] through the above two kinds of circuits, respectively.
(1)$$\left\{\begin{array}{l} P=4\frac{\sin^{2}\left(\omega_{e}t_{e}\right)e^{-\frac{\omega_{e}t_{e}}{Q_{e}}}}{{\left[1+\cos\left(\omega_{e}t_{e}\right)e^{-\frac{\omega_{e}t_{e}}{Q_{e}}}\right]}^{2}}\\ \omega_{e}t_{e}=\arctan\left(-\frac{2m}{\sqrt{\xi_{R}}}\right)+\pi \end{array}\right.$$
(2)$$\left\{\begin{array}{l} A=20\log\left(\frac{1}{1+\frac{4}{\pi }k_{m}^{2}Q_{m}\frac{1+\lambda }{1-\lambda }}\right)\\ \lambda =-\left[\tan^{2}\left(\omega_{e}t_{e}\right)+\frac{1}{2Q_{e}}\tan\left(\omega_{e}t_{e}\right)+1\right]\cos\left(\omega_{e}t_{e}\right)e^{-\frac{\omega_{e}t_{e}}{2Q_{e}}}\\ \tan\left(\omega_{e}t_{e}\right)=\frac{\pi -4k_{m}^{2}Q_{m}}{2k_{m}^{2}Q_{m}}\tau f_{e}\lambda -\frac{\pi +4k_{m}^{2}Q_{m}}{2k_{m}^{2}Q_{m}}\tau f_{e} \end{array}\right.$$

Among the equations, *ω_e_t_e_* and *Q_e_* are the inversion phase and the quality factor of the LC oscillating circuit, respectively. *F_e_* is the control force factor in the SSDEI circuit that is as a function of the voltage *V_DC_* of the injection circuit, while τ is the normalized energy injection time of this circuit. *K_m_*^2^*Q_m_* represents the merit of the piezoelectric coupling structure. Detailed derivation can be found in Refs. [20,21].

The specific results are shown in Figure 9; a circuit simulation software named Ltspice was selected to set up the piezoelectric interface circuit and define the control logics of corresponding switches. By controlling the switches in the circuit in order to make them produce different functions, the simulation results of the “suppress vibration and fill vacancy” phenomenon are ultimately obtained. Namely, when the circuit takes on the “suppress vibration” function, part of the electric energy is injected into the piezoelectric element, and this generates a braking force which could suppress the structural oscillation, resulting in a relatively high piezoelectric voltage. When the circuit takes on the “fill vacancy” function, it could harvest electrical energy generated by the piezoelectric element in the circuit at the moment; because the electricity is harvested through energy absorption, resulting a relatively low piezoelectric voltage.

## 4. Experimental Setup and Results Discussion

The designed wind–solar hybrid energy harvesting device and the performance testing platform were set up as shown in Figure 10. The hybrid energy harvesting device included a PV panel used for solar energy harvesting and a vortex-induced vibration device used for wind energy harvesting based on the direct piezoelectric effect. In addition, a low-speed wind tunnel was manufactured in order to simulate the environmental wind. The “suppress vibration and fill vacancy” algorithm was realized by the piezoelectric element and the interface control circuit. The principle of wind energy harvesting refers to the fact that when the vortex shedding frequency caused by the blunt body and the wind is close to the natural frequency of the device, large structural vibration energy will be generated. According to the above principle, a hollow cylinder made of a lightweight photosensitive resin material was designed to be affixed to the top end of the cantilever structure. This material was specially applied in a 3D printing machine, and the diameter of the cylinder could be easily selected.

Figure 11 shows the waveform of the structural vibration displacement in the wind–solar hybrid energy harvesting device under a wind speed of 2.5 m/s. This displacement waveform was attained through a laser Doppler vibrometer, simplified as LDV (OFV-505/5000, Polytec, Baden-Württemberg, Germany). In fact, the vibration displacement amplitude of the real structure at the head of blunt body exceeds 20 mm. To simplify the measurement, the test point of the LDV is in the middle of cantilever beam, a little above the piezoelectric element. According to the waveform curve, it could be seen that at this point, the vibration displacement amplitude was around 2.25 mm, and the frequency was 2.16 Hz. In order to verify the proposed wind–solar hybrid energy harvesting method, the interface control circuit and the switch control signals provided by the real-time control system (speedgoat, Bern, Switzerland) were also selected in the experimental setup.

As is shown in Figure 12, we conducted experiments in order to verify the proposed hybrid energy harvesting approach. The red curves in the figure show the vibration displacement, and the blue curves show the voltage of the piezoelectric element. The set of waveform data was deduced from an oscilloscope, and then it was replotted using MATLAB software. When it is in the suppress-vibration functional state, because the piezoelectric element is injected with electricity, the voltage is relatively high, and the amplitude waveform of displacement is nearly zero, so the structural vibration amplitude is suppressed to zero. When it is in the fill-vacancy functional state, the electric energy of the piezoelectric element is harvested to subsequent circuits due to the positive piezoelectric effect, the voltage waveform is relatively small, and the displacement waveform shows that the structure is in an oscillating state. Please note that in Figure 12a, the piezoelectric voltage waveform and simulation results are inconsistent. This is because, when the actual piezoelectric element is placed in the interface control circuit, a current drain phenomenon will occur. It is obvious when the voltage is relatively high.

Finally, the experimental demonstration of the oscillation-induced dust-clean is discussed. At the beginning, the selected clean PV plane output a maximal power of 58.56 mW. As soon as it was covered by white chalk dust, the output power under the same sunlight intensity dropped to 10.58 mW. The proposed oscillation–induced–dust–cleaning approach was then simulated. After this process, the maximal solar energy harvested power increased to 26.11 mW, all of these values are listed in Table 1. Although the value of 26.11 mW was still much lower than that of the original case, it was already 2.47 times higher than that of the dirty PV plane without the cleaning action. This experimental result clearly validates that the proposed approach indeed has the function of oscillation-induced dust-cleaning.

## 5. Conclusions

Self-powered wireless sensor nodes will be an important part of smart agriculture. Based on this, a novel wind–solar hybrid energy harvesting approach with an oscillation-induced dust-cleaning function for actual typical agricultural scenarios was thoroughly introduced in this study. This approach could compensate for the sharp oscillations in solar power generation caused by day and night alternating changes, weather changes, etc. With a specific control circuit, the approach could clean PV panels on rainy days and slow down the attenuation trend of solar power generation as much as possible maintenance-free at the same time. In addition, there is no need to maintain the panels regularly with the control circuit. Ongoing studies could aim at improving the cleaning efficiency of PV panels, since this value is still very low when compared with that in original clean PV panels. 

## Figures and Tables

**Figure 1 micromachines-14-00058-f001:**
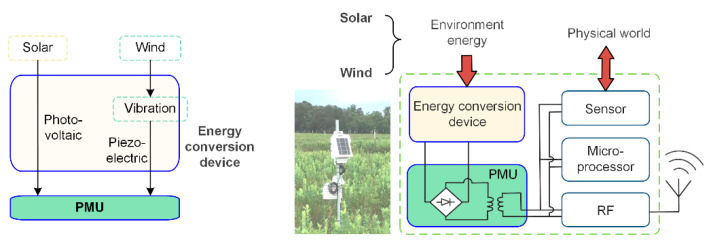
Structural chart of the operation principle of an energy conversion device and a typical autonomous sensing device.

**Figure 2 micromachines-14-00058-f002:**
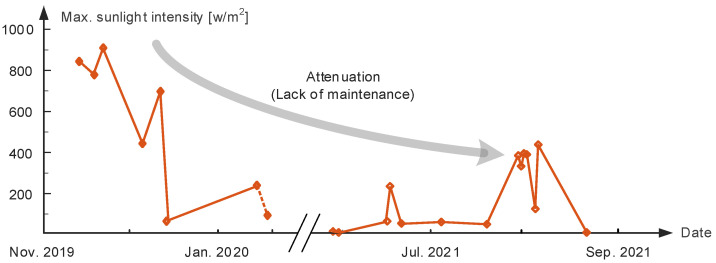
Sunlight intensity data as a function of date.

**Figure 3 micromachines-14-00058-f003:**
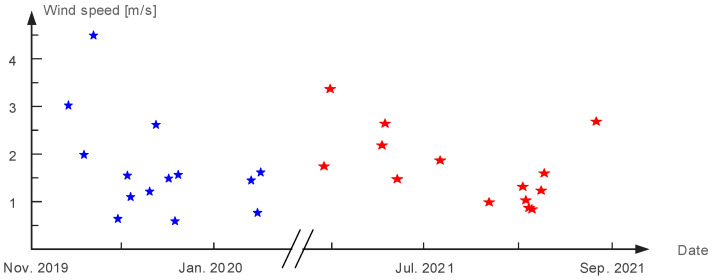
Wind speed data as a function of date.

**Figure 4 micromachines-14-00058-f004:**
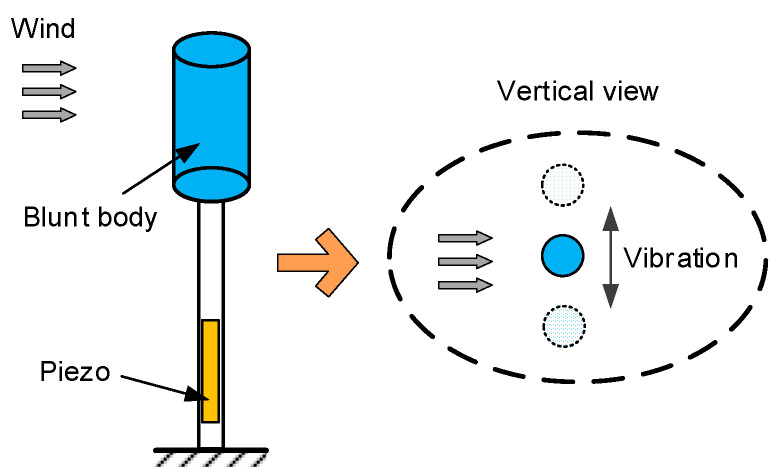
Wind-induced vibration device.

**Figure 5 micromachines-14-00058-f005:**
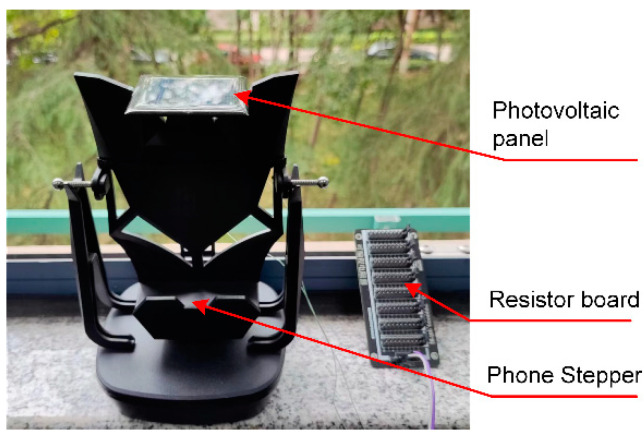
Experimental measurements of micro-power photovoltaic panels under dynamic condition.

**Figure 6 micromachines-14-00058-f006:**
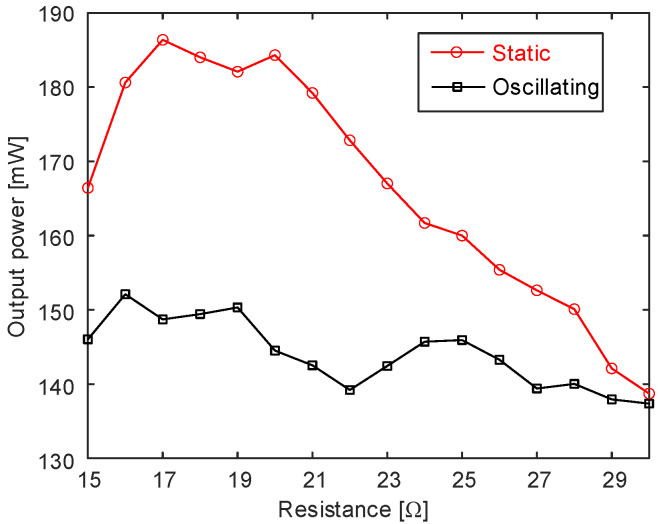
Solar panel output power with load resistance under static and dynamic conditions.

**Figure 7 micromachines-14-00058-f007:**
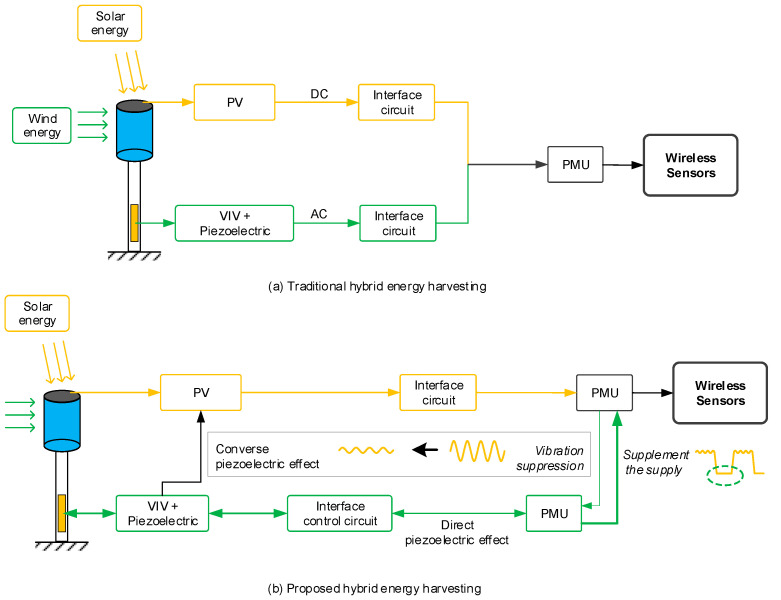
Approach comparison between the traditional and the proposed wind–solar hybrid energy harvesting methods. (**a**) Describes the process of traditional hybrid energy harvesting; (**b**) describes the process of proposed hybrid energy harvesting.

**Figure 8 micromachines-14-00058-f008:**
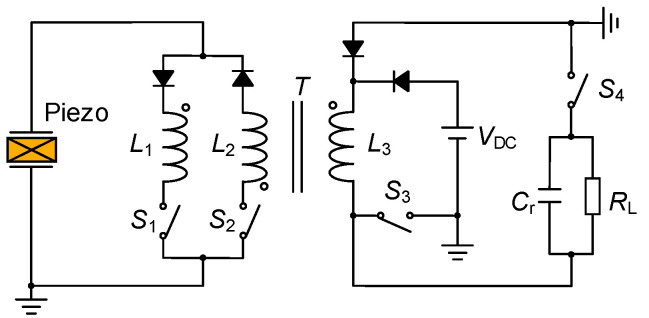
Piezoelectric interface circuit based on flyback transformer.

**Figure 9 micromachines-14-00058-f009:**
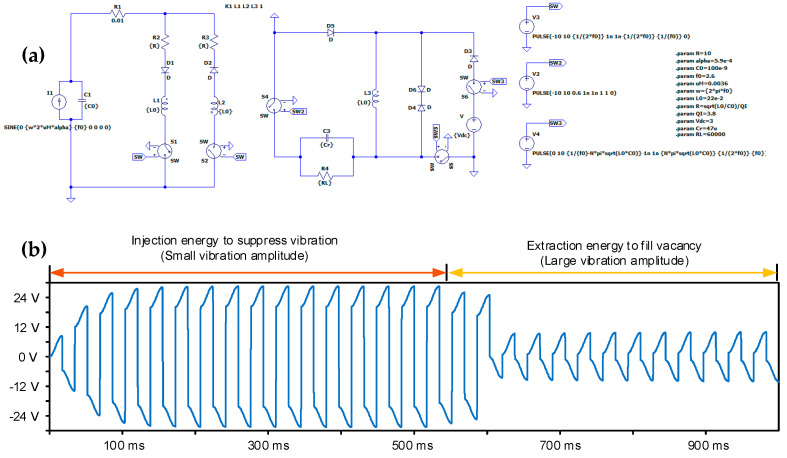
Simulation results of the “suppress vibration and fill vacancy” function using an interface circuit: (**a**) the simulation circuit in LTspice; (**b**) the results of the “suppress vibration and fill vacancy” function.

**Figure 10 micromachines-14-00058-f010:**
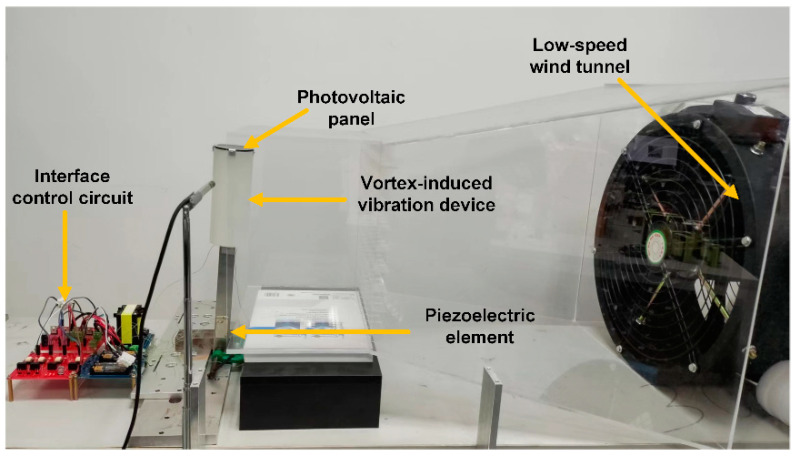
Platform of the wind-induced vibration–solar hybrid energy extraction device.

**Figure 11 micromachines-14-00058-f011:**
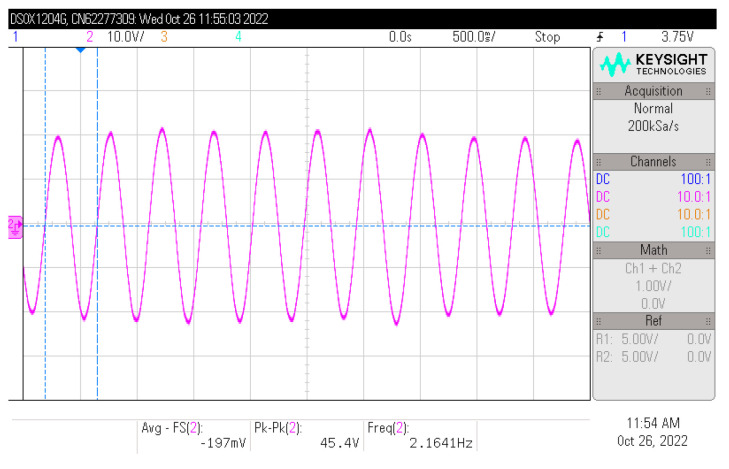
Waveform of structural vibration displacement under the wind speed of 2.5 m/s.

**Figure 12 micromachines-14-00058-f012:**
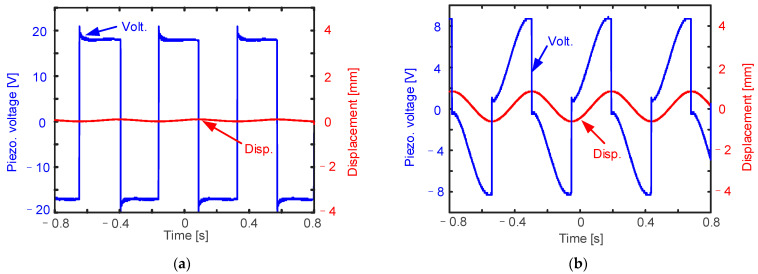
Experimental results of the “suppress vibration and fill vacancy” function: (**a**) injecting energy to suppress vibration; (**b**) extracting energy to fill vacancy.

**Table 1 micromachines-14-00058-t001:** Solar energy harvested powers under different conditions.

	Original	Dust Cover	After Cleaning
Output Power [mW]	58.56	10.58	26.11

## Data Availability

The experimental data has been provided in Figure 11 and Figure 12 (all of the wavdforms were measured by an oscillscope in the experiment).

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
