# Peer review of "A Wind-Solar Hybrid Energy Harvesting Approach Based on Wind-Induced Vibration Structure Applied in Smart Agriculture"

_micromachines, 2022, doi:10.3390/mi14010058_

Round 1

Reviewer 1 Report

I am afraid the author should receive more training on scientific writing. This is a scientific/technology paper, not propaganda. The readers expect something on scientific and technology, being specific and through. I am afraid most contents in pages 1-3 are not necessary. You should also organize the materials in a logic manner.   

Author Response

Dear Reviewer

Thank you very much for the incisive comment. We revised the manuscript once more according to this comment.

In the revised version, two key scientific results were added to support the proposed approach. The first result was the value of efficiency improvement of solar energy harvesting, the harvested power was enhanced from 10% to 15% in sunny days after using the vibration suppression algorithm. The second result was the dust cleaning efficiency of the proposed wind induced vibration approach was 2.47 times enhanced than before.

In addition, the descriptions of wind energy harvested power, the comparison of the powers, etc. were also added in the revised manuscript.

Please find the revised version of the manuscript entitled “A wind-solar hybrid energy harvesting approach based on wind-induced vibration structure applied in smart agriculture”.

We appreciate the Reviewers’ warm work and hope that the correction will meet with approval.

Sincerely yours,

Lili Xia and Yipeng Wu

Reviewer 2 Report

This manuscript developed a novel wind-solar hybrid energy harvesting approach based on wind-induced vibration structure applied in smart agriculture. A wind-induced vibration device is specially designed to generate electrical energy and/or clean the photovoltaic panel. The proposed method is innovative and the writing is good. However, a few small problems should be solved before it can be accepted to publish on Micromachines.

1. There is no need to add “novel” to the title of the manuscript;

2. Could you give the technical bottlenecks or limitations of the proposed method in the conclusion? This may provide reference suggestions for practical application scenarios.

Author Response

Please see the attachment. Thanks so much for your helpful comments.

Reviewer 3 Report

A compound device for cleaning photovoltaic panels was made in this manuscript. The description of some problems needs to be clarified. At the same time, the development and characterization of some data need to be improved. Revise and supplement according to the following issues.

1. Light and wind in fig1 as ambient energy sources. It is necessary to have a schematic diagram and a structural diagram for how the device collects wind energy.

2. Self-cleaning is mentioned in the article, and the cleaning process is expected to be described in more detail with physical pictures and videos.

3. In Fig2, the collection of the cleaned photovoltaic panels should be attached. At present, fig2 cannot explain that the reason for the drastic drop in collection efficiency is caused by dust.

4. The maximum peak value collected by the wind speed sensor has also dropped significantly. Please verify that the sensor is not affected by damage or dust.

5. The purpose of adding wind energy harvesting devices is to compensate for the energy loss of photovoltaic panels. However, the vibration caused by the wind energy harvesting device reduces the energy harvesting efficiency of the photovoltaic panel. There are two core problems. First, how much the oscillation reduces the energy harvesting efficiency of photovoltaic panels. Second, how much can the wind energy harvesting device compensate for the energy loss of photovoltaic panels? The above two problems require repeated experiments, and the results must be statistically tested and attached to the supporting literature.

6. As stated in the text, vibration is the way to clean photovoltaic panels. What is the cleaning efficiency value?

Author Response

Please see the attachments, thank you so much for your helpful comments.
